# Research on Metal Foreign Object Detection of Electric Vehicle Wireless Charging System Based on Detection Coil

**Tao Meng [1], Linlin Tan [1,\*], Ruying Zhong [1], Huiru Xie [1] and Xueliang Huang [1,2]**

[1] School of Electrical Engineering, Southeast University, Nanjing 210002, China; lmtmttt@163.com (T.M.); mirandazry@163.com (R.Z.); huiruX@seu.edu.cn (H.X.); xlhuang@seu.edu.cn (X.H.)
[2] Jiangsu Provincial Key Laboratory of Smart Grid Technology and Equipment, Nanjing 210002, China
\* Correspondence: Tanlinlin@seu.edu.cn

**Abstract:** Due to the open structure of wireless charging systems, foreign objects are inevitably involved. In the process of energy transfer, the intervention of metal foreign objects will cause the system to deviate from the normal operating point and even cause safety accidents. Therefore, metal foreign objects detection (MOD) technology is of considerable importance. In this paper, a metal foreign object detection system of a wireless power transfer (WPT) system based on the symmetrical detection coil was designed, and a MOD method based on frequency scanning is proposed. The coil size of the WPT system was designed based on SAE-J2954 standard. In addition, the suitable detection coil was installed at the transmitting coil. The system model was established in ANSYS simulation software, and 5 cm × 5 cm copper sheet and iron sheet were used as the metal to be detected. The result shows that the self-induction of the detection coil at the position of the foreign object can achieve a fluctuation of 52.46% and −34.72%, and the resonant frequency of the detection system is offset by 24.7% and −18.82%, respectively. Finally, the experimental platform of MOD system was built, and the effectiveness of the system was verified.

**Keywords:** wireless charging; electric vehicle (EV); finite element calculation; metal object detection

## 1. Introduction

The development of WPT technology is gradually maturing. In particular, the magnetic coupled resonant (MCR) transmission mode has made remarkable improvement in power level, transmission efficiency, anti-migration performance, and so on [1–4]. The MCR-WPT system transmits the energy of the transmitting coil to the receiving coil through electromagnetic induction coupling, and the resonator generates a strong alternating current magnetic field [5]. Because the charging system is open in structure, it is inevitable that there will be foreign objects. In particular, when a metallic foreign object is near the charging system, eddy currents will be induced, and a large amount of heat will be generated. As a result, part of the power transmitted by the transmitting coil is absorbed by the foreign matter, reducing the efficiency of the system, and the heat generated by the eddy currents in the metal will threaten the safety of the entire system [6–8]. In order to improve the security of WPT systems, it is necessary to identify and warn metallic foreign object before the system starts or runs.

In the research of MOD, some methods were proposed from different angles. From the perspective of characteristic parameter detection, the decrease degree of Q value caused by a metal foreign object is greater than the reduction degree of system transmission efficiency, and a MOD system based on the change of the Q value of the receiving coil was designed [9]. A foreign object detection method based on power loss was proposed, and a fast analysis algorithm for mathematical regression analysis of experimental data was designed to improve the detection accuracy [10]. A multifunctional tunnelling reluctance sensor matrix was proposed, which can detect coil alignment and metal foreign object between coils [11, 12]. However, the detection accuracy of the characteristic parameter detection method and

the power loss method cannot reach a high level, while the sensor detection method can identify the type of foreign object, but the application cost is relatively higher, and the external interference problem is difficult to deal with. With its flexibility and adaptability, the coil detection method has been extensively focused on at present. This method is mainly based on the characteristic parameter method, which converts the influence of metal foreign objects on the parameters of the energy coil into the influence of metal foreign objects on the parameters of the detection coil, so as to enlarge the parameter change multiple and improve the sensitivity of detection. A dual-purpose non-overlapping coil group was proposed for metal object detection and electric vehicle position detection, the metal foreign object was determined by the inductive voltage of the detection coil group and the coil group did not produce any power loss [13,14]. A double-layer symmetrical detection coil considering magnetic field characteristics was proposed. The voltage difference between each detection coil group was used to detect the position of foreign object, which can reduce the design complexity. However, this scheme had a blind spot in detection [15]. In order to eliminate the blind area, a symmetrical induction coil design scheme is proposed, which eliminates the detection blind area of a MOD system by staggered arrangement of detection coils [16]. Reference [17] proposed a MOD system with fuzzy resonant circuit. The resonant circuit adopted the parallel mode, and the driving power frequency was raised to 1 MHz to reduce the harmonic magnetic field interference of the EV-WPT system. Reference [18] proposed a multilayer detection coil layout that not only covered the entire charging area to eliminate blind areas, but also could be separated from the transmitting and receiving coils to minimize the impact on the magnetic field of the energy-transmitting coil. However, the current detection coil methods are mainly at the level of whether it can be detected or not. Some systems can detect the position of foreign objects but cannot realize the recognition of the type. The comparison of MOD methods for the EV-WPT system is given in Table 1.

**Table 1.** Comparison of FOD technologies.

| Method | Advantages | Disadvantages | Suitable Scene |
|---|---|---|---|
| Parameter detection | No additional equipment required | Low sensitivity | Low power level, low accuracy |
| Sensor detection | Wide recognition range | High cost, vulnerable to external interference | Regardless of the cost |
| Coil detection | Low cost, high sensitivity, multifunctional | Occupied space, subject to coil interference | High power level, high accuracy |

In order to identify the position and type of foreign objects, this paper adopts the method of detecting coil, establishes the suitable detecting coil arrangement, and designs the detection circuit to achieve higher detection accuracy. A method based on frequency scanning is proposed to realize the recognition of metal types. The MOD system proposed in this paper is beneficial to improve the security of the EV-WPT system.

## 2. Influence Analysis of Metal Foreign Objects

Different metal types have different effects when they are near the WPT system. When the non-ferromagnetic metal is near the transmitting coil, the impedance parameters of the transmitting coil will be affected by the eddy current effect of the metal [19]. When the ferromagnetic metal is close to the coil, the magnetic effect will also affect the coil parameters in addition to the eddy current effect [20]. This section will model the WPT system based on metal foreign objects and analyze the influence of different types of metals on coil parameters theoretically.

### 2.1. Influence Characteristics of Non-Ferromagnetic Metals

When there is a metal foreign object between the coils, due to the relatively high conductivity of the metal material, the alternating magnetic field between the primary

and secondary coils of the WPT system induces current on the metal material and forms a closed loop, as referenced as Figure 1. At the same time, the eddy currents in the metal plate also create a magnetic field to counteract the magnetic field between the coils, thus changing the coil parameters of the system.

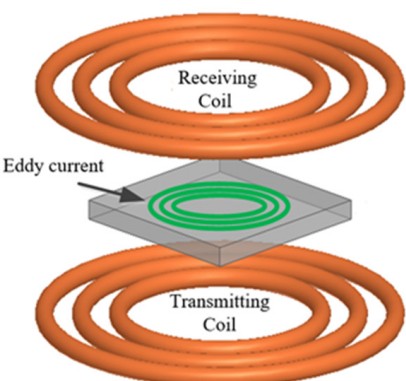

**Figure 1.** Metal foreign object and eddy current.

According to the loosely-coupled transformer model, the eddy current on non-ferromagnetic metals can be equivalent to series equivalent circuits of inductors and resistors, and is magnetically coupled to the coils. Therefore, the equivalent circuit of the EV-WPT system in the presence of non-ferromagnetic metal objects is shown in Figure 2, where $L_m$ and $R_m$ are the metal's equivalent self-inductance and equivalent internal resistance, respectively, $I_m$ is the metal's equivalent eddy current, and $M_{pm}$ and $M_{sm}$ are the mutual inductance between the metal and the coil, respectively.

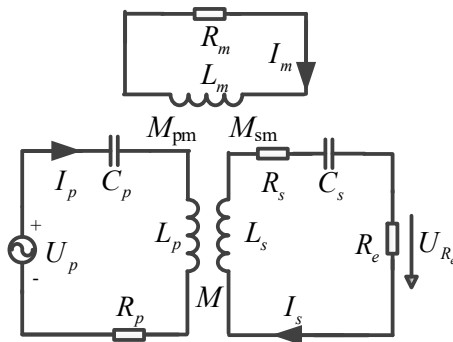

**Figure 2.** Equivalent circuit.

According to the KVL equation:

$$\begin{cases} \dot{U}_P = Z_p \dot{I}_p - j\omega_s M \dot{I}_s - j\omega_s M_{pm} \dot{I}_m \\ 0 = Z_s \dot{I}_s - j\omega_s M \dot{I}_p - j\omega_s M_{sm} \dot{I}_m \\ 0 = Z_m \dot{I}_m - j\omega_s M_{pm} \dot{I}_p + j\omega_s M_{sm} \dot{I}_s \end{cases} \tag{1}$$

When there is a metal foreign object between the coils, due to the relatively high conductivity of the metal material, the alternating magnetic field between the primary and secondary coils of the WPT system induces current on the metal material and forms a closed loop, as referenced as Figure 1. At the same time, the eddy currents in the metal plate also create a magnetic field to counteract the magnetic field between the coils, thus changing the coil parameters of the system.

The equivalent mutual inductance $M'$, inductance ($L_P'$, $L_S'$) and equivalent resistance ($R_P'$, $R_S'$) of the EV-WPT system in the presence of non-ferromagnetic metals are obtained:

$$
\begin{cases}
L_p' = L_p - \dfrac{L_m \omega_s^3 M_{pm}^2}{R_m^2 + \omega_s^2 L_m^2} \\
L_s' = L_s - \dfrac{L_m \omega_s^3 M_{sm}^2}{R_m^2 + \omega_s^2 L_m^2}
\end{cases}
\tag{2}
$$

$$
\begin{cases}
R_p' = R_p + \dfrac{R_m \omega_s^2 M_{pm}^2}{R_m^2 + \omega_s^2 L_m^2} \\
R_s' = R_s + \dfrac{R_m \omega_s^2 M_{sm}^2}{R_m^2 + \omega_s^2 L_m^2}
\end{cases}
\tag{3}
$$

When non-ferromagnetic metals are present in the EV-WPT system, the eddy currents in the metal will reduce the equivalent mutual inductance, reduce equivalent self-inductance, and increase the equivalent internal resistance of the coil. This is because the eddy current will generate an inverted magnetic field to offset the magnetic field of the transmission coil, which will lead to the decrease of the self-inductance of the primary and secondary coil, and further increase the resonance frequency point of the system.

### 2.2. Influence Characteristics of Ferromagnetic Metals

When the ferromagnetic metal is close to the system, the relative permeability in space changes from $\mu_0$ to $\mu_r$. The magnetic effect increases the inductance of the transmitting coil, and the inductance that increases the magnetization effect of the ferromagnetic metal is $L_m$. The equivalent impedance of the transmitting coil is:

$$
Z = R_p + \frac{\omega^2 M^2 R_s}{R_s + \omega^2 L_s^2} - \frac{\omega^2 M^2 L_s}{R_s + \omega^2 L_s^2} + j\omega L_m
\tag{4}
$$

Compared with non-ferromagnetic metal materials, ferromagnetic metal materials have a relative permeability of much more than 1, which means that a ferromagnetic metal foreign object can not only induce eddy currents, but also produce a magnetic field in the same direction as the external magnetic field because of the magnetic effect. The influence of ferromagnetic metal material on self-induction of single-turn circular coil is analyzed. The eddy current effect of the foreign object of ferromagnetic metal will increase the resistance of the coil, and the self-inductance change of the coil needs to consider both eddy current effect and magnetic effect, which mainly depends on the relative strength of magnetic effect and eddy current effect.

### 3. MOD System Design

Based on the analysis of the influence characteristics of metal foreign objects, it can be seen that different types of metal have different effects on the self-inductance parameters of the coil, which will be the key basis for MOD. Therefore, a MOD system based on detection coil array is proposed in this paper, and its schematic diagram is shown in Figure 3. The detection coil array can not only improve the identification accuracy, but also can identify the position of foreign object without the existence of blind spots.

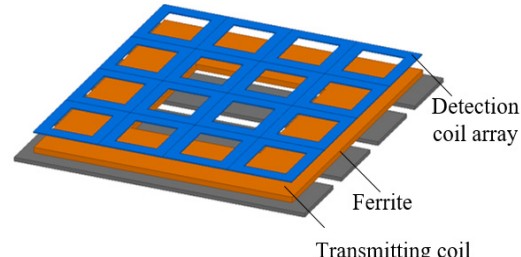

**Figure 3.** WPT system with detection coil array.

### 3.1. Detection Circuit Design

The detection circuit is shown in Figure 4, which is divided into the coil module, detection module and output module. $L_1 \sim L_n$ are the self-induction of unit detection coils, $C_1 \sim C_n$ are the matching resonant capacitors. In this method, the position of the metal is determined by the loss of resonance due to the offset of the inductance parameter of the detection coil, and the type can be identified by obtaining the offset of resonant point by sweeping frequency.

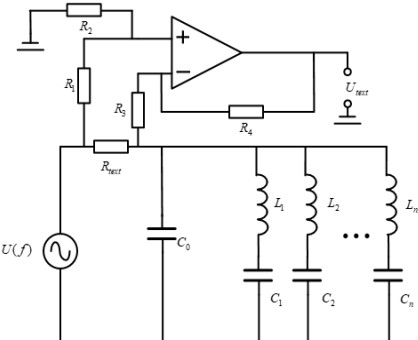

**Figure 4.** Equivalent circuit of MOD.

Resonant capacitor parameters need to be matched in the absence of a metal foreign object, as follows:

$$\omega_0 = 2\pi f_0 = \frac{1}{\sqrt{L_1 C_1}} = \frac{1}{\sqrt{L_2 C_2}} = \cdots = \frac{1}{\sqrt{L_n C_n}} \tag{5}$$

When there is a metal foreign object on the detection coil, the equivalent inductance can be expressed as:

$$L_1' = L_1 + \Delta L \tag{6}$$

The working frequency of the detection system is set as $f_0$, and the input impedance of the main circuit module in the presence of a metal foreign object can be expressed as:

$$Z_{in} = R_{test} - j\omega_0 L_1 (1 + \frac{L_1}{\Delta L_1}) \tag{7}$$

The effective values of the voltage at both ends of the test resistance $R_{test}$ can be expressed as:

$$U_{AB} = \frac{U}{|Z_{in}|} = \frac{U}{\sqrt{R_{test}^2 + \omega_0^2 L_1^2 (1 + \frac{L_1}{\Delta L})^2}} \tag{8}$$

While amplifying $U_{AB}$, it is necessary to ensure that the amplitude of the output voltage is less than the supply voltage of the op-amp, so that it can work in a linear workspace. The voltage output by the sampling module is sent to the AD sampling port through the peak detection module, which can be written as:

$$U_{test} = \sqrt{2} U_o \cdot = \frac{R_2 R_7}{R_1 (R_6 + R_7)} \cdot \frac{\sqrt{2} U}{\sqrt{R_{test}^2 + \omega_0^2 L_1^2 (1 + \frac{L_1}{\Delta L})^2}} \tag{9}$$

According to Equation (9), when the parameters are determined, the variation of the inductance parameters of the detection coil increases, the module of the input impedance decreases, and the detection voltage $U_{test}$ increases accordingly. Actually, outside interference and fluctuations in the WPT system can also cause $U_{test}$ to fluctuate slightly, which can lead to misjudgements in the MOD system. Therefore, it is necessary to set threshold voltage ($U_{thre}$) to improve the anti-jamming ability of the system. The value of $U_{thre}$ needs

to take into account the external disturbance and the sensitivity of the detection coil. When $U_{test} > U_{thre}$, it can be determined that there is a metal foreign object on the detection coil, the formula can be written as:

$$\frac{\sqrt{\left[\frac{R_2 R_7}{R_1(R_6+R_7)} \cdot \frac{\sqrt{2}U}{U_{thre}}\right]^2 - R_{test}^2}}{\omega_0^2 L_1^2} - 1 > \frac{L_1}{\Delta L} \tag{10}$$

Only the detection coil where the metal object is located receives the most interference and the $U_{test}$ value is the largest, so the location of the metal object can be detected. The above theoretical analysis is used to identify the location of a metal foreign object, and then the type of metal is determined by means of frequency sweep. When a metal foreign object is present in the detection coil, in order to make the detection system resonate, the frequency of the power supply shall be written as:

$$\omega_1 = 2\pi f_1 = \frac{1}{\sqrt{L_1' C_1}} = \frac{1}{\sqrt{(L_1 + \Delta L)C_1}} \tag{11}$$

Thus, the resonant frequency offset after the presence of a metal foreign object is:

$$\Delta f = \frac{|\omega_1 - \omega_0|}{2\pi} = \frac{1}{2\pi\sqrt{L_1 C_1}} \left| \frac{1}{\sqrt{1 + \frac{\Delta L_1}{L_1}}} - 1 \right| = f_0 \cdot \left| \frac{1}{\sqrt{1 + \frac{\Delta L_1}{L_1}}} - 1 \right|$$

$$= \begin{cases} f_0 \left( \frac{1}{\sqrt{1 + \frac{\Delta L_1}{L_1}}} - 1 \right) & (\Delta L_1 < 0) \\ f_0 \left( 1 - \frac{1}{\sqrt{1 + \frac{\Delta L_1}{L_1}}} \right) & (\Delta L_1 > 0) \end{cases} \tag{12}$$

According to (12), the change of self-induction of detection coil caused by different types of metal foreign objects can be converted into the change of resonant frequency, so the type of metal foreign object can be identified.

### 3.2. Detection Coil Design

The next step is to design the magnetic circuit module in the MOD system. The design criteria mainly include the sensitivity to respond to foreign objects and the economy of detection coils. Response sensitivity refers to the reaction degree of the detection coil after the metal foreign object enters the detection coil, which is specifically reflected in the rate of change of self-induction, while the economy of the detection coil is reflected in the number of detection coils. The two indexes are mutually restricted.

In the design of detection coils, it is necessary to determine the optimal number of detection coils. Considering the symmetry of the EV-WPT system selected in this paper, the number of detection coils is set as 4, 9, 16 and 25, respectively. In the simulation, the number of turns of detection coils is set as 16, and the copper sheet with a side length of 5 cm is used as a metal foreign object, so the self-inductance changes under a different number of detection coils can be obtained (as shown in Table 2).

**Table 2.** The change of self-inductance under different number of detection coils.

| Number | Self-Inductance (Non-Foreign Object)/μH | Self-Inductance (Foreign Object)/μH | Change in Self-Induction/μH | Rate of Change |
|--------|------------------------------------------|--------------------------------------|------------------------------|----------------|
| 4 | 124 | 122.14 | 1.86 | 1.50% |
| 9 | 58.135 | 54.14 | 3.995 | 6.87% |
| 16 | 28.80 | 23.57 | 5.23 | 18.16% |
| 25 | 15 | 10.797 | 4.203 | 28.02% |

According to Table 2, the change rate of self-inductance increases with the increase of the number of detection coils. By comparison, when the number of detection coils increases from 9 to 16, and the self-inductance change rate increases by 11.29%, then the self-inductance change rate of unit coil increases by about 1.62%. However, when the number of detection coils increases from 16 to 25, the self-inductance change rate of detection coil only increases by 9.86% and the self-inductance change rate of unit coil only increases by about 1.09%. With the increase of the number of detection coils, the self-inductance change rate of detection coils first increases and then decreases. The more the number of detection coils, the higher the cost of detection coils. Therefore, considering the increase of self-inductance change rate and the restriction of cost, the number of detection coils of 16 can achieve the best balance of response sensitivity and economy.

## 4. Simulation and Experiment

This section will conduct simulation analysis based on the MOD system designed in Section 3. Common metal foreign objects mainly include cans, copper blocks, steel sheets, steel velvet, nails, etc. In order to make this method universal, this section mainly analyzes two typical metal foreign objects of copper block and steel sheet, and their parameters are shown in Table 3.

**Table 3.** The parameters of metal foreign object.

| Material | Size | Parameter | Value |
|---|---|---|---|
| Copper | 50 mm × 75 mm × 5 mm | Relative permeability<br>Conductivity | 0.999991<br>58,000,000 S/m |
| Steel | 50 mm × 75 mm × 5 mm | Relative permeability<br>Conductivity | 300<br>2,000,000 S/m |

In order to increase the magnetic field gathering capacity between primary and secondary coils and thus increase the coupling coefficient between primary and secondary coils, a ferrite plate is usually placed below the coil in the actual EV-WPT system. Therefore, in order to model the actual working conditions of the EV-WPT system as much as possible, the coil system (as shown in Figure 5) is established in ANSYS software. The coil system has a matching ferrite to enhance coupling, and the transmitting and receiving coils are of the same type. Meanwhile, the parameters of the coil system are listed in Table 4.

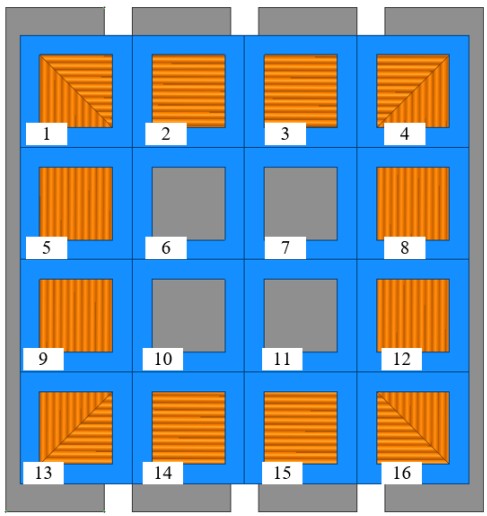

**Figure 5.** Symmetrically shaped detector coil array and WPT system coil.

**Table 4.** System parameters.

| Object | Parameter | Value | Object | Parameter | Value |
|--------|-----------|-------|--------|-----------|-------|
| Coil | Inner diameter | 100 mm | Ferrit | Length | 400 mm |
| | Outer diameter | 200 mm | | Width | 400 mm |
| | Turns | 10 | | Thickness | 5 mm |
| | Wire diameter | 2.5 mm | | Distance from coil | 1 mm |
| | Transmission distance | 190 mm | | | |

Considering the symmetry of the detection coil array in both X and Y axes, only the self-inductance change parameters of the metal foreign object at 1/4 of the transmitting coil need to be simulated. The simulation model is shown in Figure 6. When the metal foreign object moves within the range of 1/4 of the transmitting coil, the self-inductance variation of the detection coil on each detection coil is shown in Figure 7.

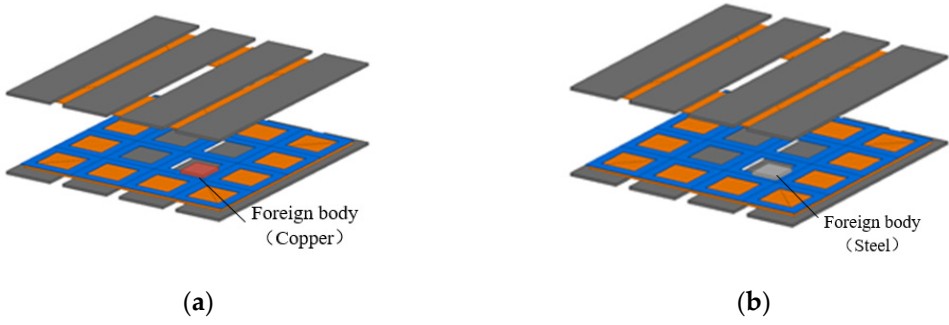

(**a**)　　　　　　　　　　　　　　　　(**b**)

**Figure 6.** Simulation model. (**a**) Copper sheet as a metal foreign object; (**b**) Steel sheet as a metal foreign object.

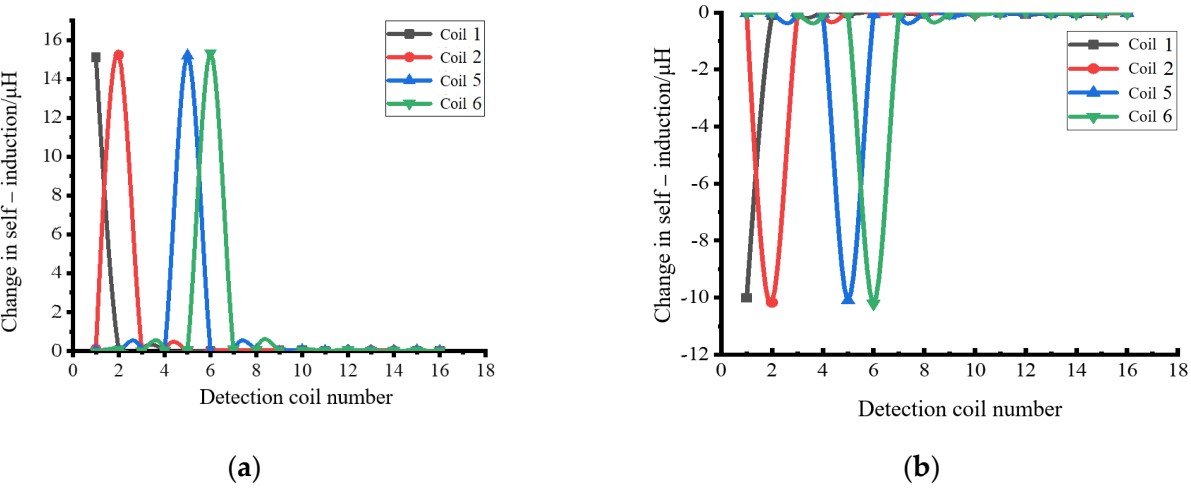

(**a**)　　　　　　　　　　　　　　　　(**b**)

**Figure 7.** The self-inductance change of the detection coil. (**a**) Self-inductance curve under the copper sheet. (**b**) Self-inductance curve under the steel sheet.

Frequency scanning is performed on the detection coil where the metal is located, and the impedance curve of the detection coil is shown in Figure 8.

The simulation result shows that the self-inductance parameters of the detection coil fluctuate under the metal foreign object, while the coils at other positions are not affected. The influence of copper and iron on the detection coil is about 15.11 µH and −10 µH, and the self-inductance change rates are 52.46% and −34.72%, respectively. As shown in Figure 8, the resonant frequency offset of the detected coil at the copper sheet is 24.7%, while that of the steel is −18.82%. Because the material of the copper block is

non-ferromagnetic material, the eddy current effect reduces the self-induction value of the detection coil and increases the resonant frequency. However, due to the magnetic effect of the steel block, the self-induction value of the detection coil increases and the resonant frequency decreases. The simulation results are consistent with the theoretical analysis in Section 2.

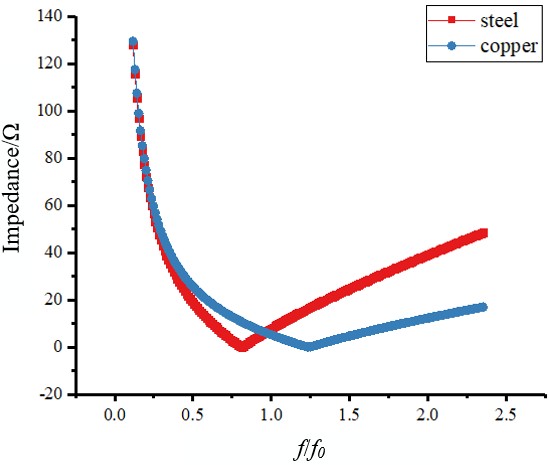

**Figure 8.** Frequency sweep curve.

In the experimental aspect, the coil model of the WPT system was built and the detection coil array for metal foreign object detection was wound. Similarly, considering the symmetrical structure of the coil, a detection coil was configured in a quarter area of the primary side coil, and a matching detection circuit was established. The overall experimental platform is shown in Figure 9. The platform mainly includes the energy transfer coil of the EV-WPT system, detection coil, harmonic compensation capacitor, detection circuit auxiliary module and DSP detection and control module.

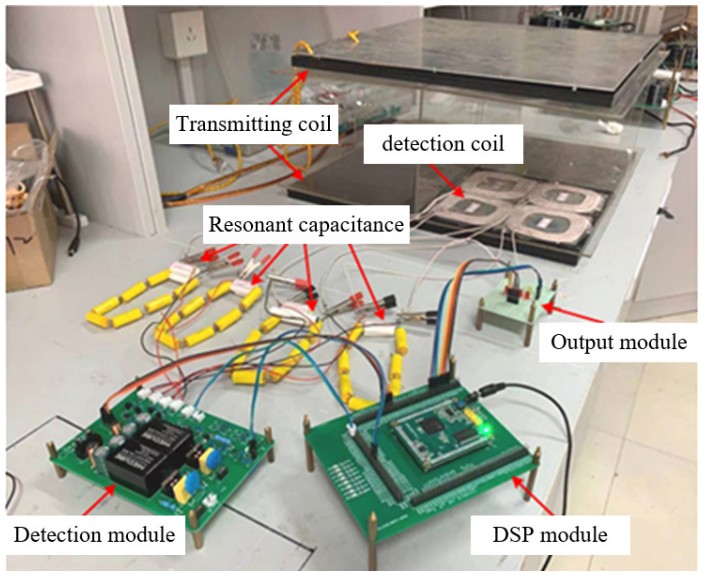

**Figure 9.** MOD experimental platform.

In this experiment platform, TMS320F28377DSP processor is used as the sinusoidal signal generator circuit in the experiment. Considering the limitations of DSP clock frequency and DA conversion chip, the sinusoidal wave of 41.7 kHz is intended to be used as the original resonant frequency of the MOD system. The parameters of the detection coil and resonant compensation capacitor in the absence of foreign object are listed in Table 5.

**Table 5.** Detection coil and resonant compensation capacitance parameters.

| Parameter | Coil 1 | Coil 2 | Coil 3 | Coil 4 |
|---|---|---|---|---|
| Self-induction/$\mu$H | 20.15 | 19.8 | 20.7 | 20 |
| Resistance/m$\Omega$ | 94 | 90 | 95.5 | 97 |
| Resonant capacitance/$\mu$F | 0.72 | 0.74 | 0.70 | 0.73 |
| Size | 10.5 cm $\times$ 10.5 cm | | | |
| Turns | 12 | | | |
| Wire diameter | 2 mm | | | |

In addition, the sinusoidal step wave generated by DSP is a unipolar waveform, which requires a reverse summations circuit to make it become a bipolar output waveform. By filtering and amplifying the bipolar output waveform, the sinusoidal waveform needed by the MOD system can be obtained. The sinusoidal step wave generated by DSP is shown in Figure 10a, and the bipolar sine wave obtained after filtering and amplification is shown in Figure 10b.

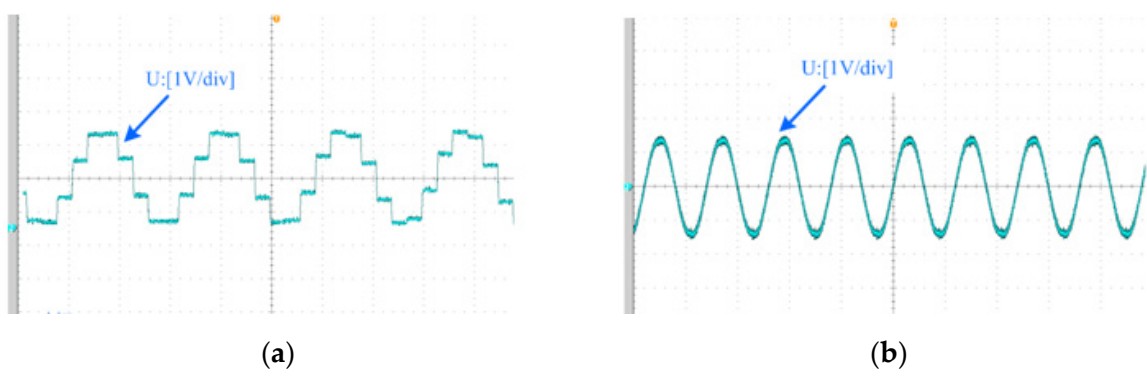

(**a**)        (**b**)

**Figure 10.** The power waveform of MOD system. (**a**) Unipolar stepped waves; (**b**) Bipolar sine waves.

In this experiment, copper sheet and steel sheet were selected as the metal foreign objects to be detected. Copper can be regarded as commonly used non-ferromagnetic metal materials, while steel can be regarded as ferromagnetic metal materials, as shown in Figure 11. Their dimensions are given in Table 6.

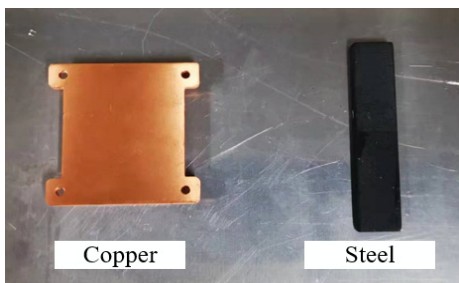

**Figure 11.** Metal foreign object.

**Table 6.** Metal foreign object size.

| Material | Magnetic Permeability | Size |
|---|---|---|
| Copper | 1 | 50 mm $\times$ 50 mm $\times$ 4 mm |
| Steel | 400 | 69 mm $\times$ 14 mm $\times$ 4 mm |

The LCR instrument was used to measure the changes of self-inductance of the metallic foreign object at different positions of the detection coil, and the range of self-inductance

changes and the resonant frequency range under the metal foreign object are listed in Table 7.

**Table 7.** The offset of self-inductance and resonance frequency.

| | Copper Sheet | | Steel Sheet | |
|---|---|---|---|---|
| | Self-Induction/µH | Resonant Frequency/kHz | Self-Induction/µH | Resonant Frequency/kHz |
| Coil 1 | 17.8–18.9 | 42.7–44.4 | 22.5–26.4 | 36.1–39.5 |
| Coil 2 | 16.9–18.5 | 43.1–45.5 | 22.2–25.9 | 36.5–39.7 |
| Coil 3 | 18.3–19.0 | 42.6–43.8 | 22.7–27.3 | 35.5–39.3 |
| Coil 4 | 17.8–18.5 | 43.1–44.4 | 22.5–26.6 | 36.0–39.5 |

According to Table 7, the self-inductance of each detection coil caused by copper sheet is offset by −10.2–15.6%, and the resonance frequency is offset by 2.4–9.1%. The self-inductance of the detection coil caused by steel sheet is offset by 12.2–30.8%, and the resonance frequency is offset by −5.3–13.6%. A metal foreign object will cause the detection system to lose resonance, resulting in a voltage mutation on the detection resistance. The waveform of the detected voltage under the copper sheet is shown in Figure 12.

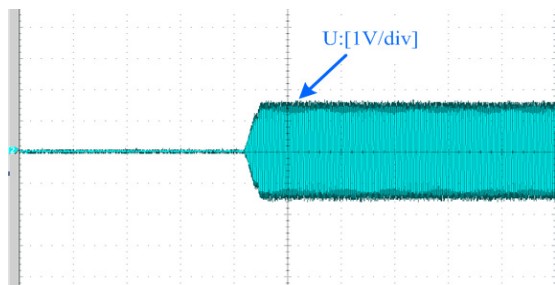

**Figure 12.** The waveform of the detected voltage under the copper sheet.

## 5. Conclusions

First, according to the demagnetization effect of eddy current and the magnetization effect of ferromagnetic materials, the influence of different metal materials of the electrical parameters of the EV-WPT system was obtained through theoretical analysis, and the influence law of coil self-induction and internal resistance was summarized. Second, in order to accurately identify the position and type of metal foreign object, a symmetrical array of detection coils was designed, and a method of MOD based on frequency scanning was proposed, and the number of detection coils suitable for electric vehicles was obtained. The proposed scheme is verified by simulation and experiment. The simulation result shows that the influence of copper and iron on the detection coil is about 15.11 µH and −10 µH, and the self-inductance change rates are 52.46% and −34.72%, respectively. The resonant frequency offset of the detected coil at the copper sheet is 24.7%, while that of the steel is −18.82%. Finally, an experimental platform for MOD-WPT system was built, and the effectiveness of the MOD scheme designed in this paper was verified through experiment.

**Author Contributions:** Conceptualization, T.M. and L.T.; methodology, T.M.; software, R.Z.; validation, L.T.; formal analysis, H.X.; investigation, T.M.; resources, T.M.; data curation, L.T.; writing—original draft preparation, L.T.; writing—review and editing, T.M.; visualization, L.T.; supervision, X.H.; project administration, T.M.; funding acquisition, T.M. and L.T. All authors have read and agreed to the published version of the manuscript.

**Funding:** This work was supported in part by the State Grid Corporation of China Technology Project "Research on Wired and Wireless Integrated Charging Technology of Electric Vehicle" (SGJSSZ00KJJS2100924).

**Institutional Review Board Statement:** Not applicable.

**Informed Consent Statement:** Not applicable.

**Data Availability Statement:** Not applicable.

**Conflicts of Interest:** The authors declare no conflict of interest.

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
