# Peer review of "Research on Metal Foreign Object Detection of Electric Vehicle Wireless Charging System Based on Detection Coil"

_wevj, doi:10.3390/wevj12040203_

Round 1

Reviewer 1 Report

This is an interesting subject for the MOD of WPT systems.

However, the reviewer has several questions and suggestions as follows

  1. The parameters in equation 1 and figure 2. are not completely matched.  The authors should improve the consistency of parameters in the article.
  2. There are the same titles of "Influence Characteristics of Non-Ferromagnetic Metals" at section 2.1 and 2.2. The authors should take care of the article's structure.
  3. The key point of the article is detection of Utest, and comparing this value with Uthre. However, the reviewer can not find the information of Uthre anywhere. What is the Uthre?
  4. The proposed idea is to detect the intervention of a foreign object, but can we also know the location of the intervention? It is not described in the paper.
  5. The proposed method detects the detection of a foreign object through a frequency sweep. It would be nice to have a waveform that can describe the frequency sweep and the resulting detction in the experimental waveform.

Author Response

Dear Reviewers:

Thank you for your comments concerning our manuscript entitled “Research on Metal Foreign Object Detection of Electric Vehicle Wireless Charging System Based on Detection Coil” (wevj-1396357). Those comments are all valuable and very helpful for revising and improving our paper, as well as the important guiding significance to our researches. We have studied comments carefully and have made correction which we hope meet with approval. Revised portion are marked in red in the paper. The main corrections in the paper and the responds to your comments are as flowing:

  1. The parameters in equation 1 and figure 2 are not completely matched. The authors should improve the consistency of parameters in the article.

Response: We are very sorry for the writing mistakes and formula errors caused by our carelessness. In the revised version, we have made significant efforts to remove the mistakes and errors. Modified equation 1 and figure 2 are given in the revised version (line107-line111). For the consistency of all parameters and the correctness of the formula, we made a careful check again.

  1. There are the same titles of "Influence Characteristics of Non-Ferromagnetic Metals" at section 2.1 and 2.2. The authors should take care of the article's structure.

Response: We are sorry for the writing mistakes caused by our carelessness again. Contrast with section 2.1, the title of section 2.2 should be “Influence Characteristics of ferromagnetic Metals”. The content of section 2.2 is also modeled and analyzed around ferromagnetic metals. Revised title is given in the revised version (line125).

  1. Response to comment: The key point of the article is detection of Utest, and comparing this value with Uthre. However, the reviewer cannot find the information of Uthre anywhere. What is the Uthre?

Response: It is really true as you suggested that we have omitted a detailed description of Uthre. Please allow us to elaborate on Uthre: According to Equation 9, the offset of the inductance parameters of the detection coil can be reflected by Utest. Actually, outside interference and fluctuations in the WPT system can also cause Utest to fluctuate slightly, which can cause the MOD system to misjudge. In the experiment, we found that when the working state of the system changed, the disturbance of the magnetic field would make Utest value not be 0(about 0.2V), even if there were no foreign object in WPT system. Therefore, it is necessary to set the threshold voltage (Uthre) to improve the anti-jamming ability of the system. As long as the value of Uthre is greater than the influence of the system and small external disturbances, and less than the influence of metal foreign objects on the coil, the detection of the MOD system can be realized and misjudgment can be avoided. The description of Uthre is given in revised version (line171-177). 

  1. The proposed idea is to detect the intervention of a foreign object, but can we also know the location of the intervention? It is not described in the paper.

Response: We are sorry that we did not clearly describe the steps of the MOD system. The first step in the MOD system is to locate the intervention. The structure of detection coil is shown in blue in Fig.6, because of the different distance between the metal foreign object and each detection coil, the influence on it is different. The influence of intervention on coil self-induction at different positions is shown in fig.7, It can be seen that only the detection coil where the metal object is located receives the most interference, so the location of the metal object can be detected. In the second step, after determining the position of the intervention, the resonance point can be determined by sweeping the frequency of the detection coil where the intervention is located, and the type of metal can be known. As you suggested that we have strengthened the description of the first step in the revised version (line178-180).

  1. Response to comment: The proposed method detects the detection of a foreign object through a frequency sweep. It would be nice to have a waveform that can describe the frequency sweep and the resulting detection in the experimental waveform.

Response: As you suggested that it would be better to have a waveform that can describe the frequency sweep in the experimental waveform. Unfortunately, our lab's impedance analyzer, which scans the coils in frequency, is broken, and new equipment is still on order. Therefore, we had to change the frequency of the power source measurement one by one, and draw the frequency offset plot instead of the scan waveform. In this way, the correctness of the proposed frequency scanning method can also be demonstrated. But in the future, we will present the results as you suggest and hope you may understand our difficulty at this stage.

All the errors you picked and recommendations you proposed are greatly helpful for us to polish our manuscript.

Thank you and best regards.

Corresponding author: Linlin Tan

Reviewer 2 Report

The paper deals with the efforts to  propose a metal foreign object detection (MOD) system of wireless power transfer (WPT) system based on the symmetrical detection coil and a MOD method based on frequency scanning.

The paper is basically technically interesting. However, there are some issues which must be improved and corrected by the authors. 

  1. In the abstract, also mention brief background of the study to show the novelty and importance of the study.
  2. In introduction section, describe in detail the background of the study, correlated previous works and literature study, and novelty of the study (including the difference with other previous works and literature). Currently, the introduction is very simple, and the readers cannot measure and understand the novelty and importance of the works. 
  3. Further literature study and references must be added significantly. It seems that the authors didn't perform a thorough literature study. Minimally about double of the current number of references are required. 
  4. There are many references which are lumped together without providing sufficient description for each of them. Provide a short description for each used reference, hence, the readers can understand and justify the contents of each reference. 
  5. The English must be improved significantly. There are many grammatical errors throughout the manuscript. 
  6. Check carefully the correct way to write the unit. In addition, provide a space between the numerical value and unit. 

Author Response

Dear Reviewers:

Thank you for your comments concerning our manuscript entitled “Research on Metal Foreign Object Detection of Electric Vehicle Wireless Charging System Based on Detection Coil” (wevj-1396357). Those comments are all valuable and very helpful for revising and improving our paper, as well as the important guiding significance to our researches. We have studied comments carefully and have made correction which we hope meet with approval. Revised portion are marked in red in the paper. The main corrections in the paper and the responds to your comments are as flowing:

  1. In the abstract, also mention brief background of the study to show the novelty and importance of the study.

Response: We are very sorry for our negligence of the background in the abstract. We have carried out further research on related study and added a brief background introduction to the abstract. Revised abstract is given in the revised version (line10-13).

  1. In introduction section, describe in detail the background of the study, correlated previous works and literature study, and novelty of the study (including the difference with other previous works and literature). Currently, the introduction is very simple, and the readers cannot measure and understand the novelty and importance of the works.

Response: Thank you for your suggestions on the introduction section. We have made correction according to your comments. The detailed background of the topic was given in the first paragraph (line26-38). Further research on foreign object detection methods is carried out, the characteristics of different methods are compared(line47-51), and the research status of coil detection methods is described in detail(line62-73). At the end of the introduction, compared with the previous work and literature, the innovation and significance of the work of this paper are proposed(line79-83).

  1. Further literature study and references must be added significantly. It seems that the authors didn't perform a thorough literature study. Minimally about double of the current number of references are required.

Response: Thank you for your suggestions. Because our main work is to innovate in the detection coil method, we have supplemented the references in this field for your comments(line62-73). Because the literature on coil detection methods is most closely related to our research, the number of references in this area has doubled. Referring to similar articles in this journal, most of the references are about 15, and only the most representative references in this field are selected for the consideration of the length of the paper. Hope you may understand our difficulty at this stage.

  1. There are many references which are lumped together without providing sufficient description for each of them. Provide a short description for each used reference, hence, the readers can understand and justify the contents of each reference.

Response: According to your suggestions, we have supplemented the literature in the field of detection coils (Line57; Line62-73). Considering the overall structure of the paper, we mainly supplement the literature that can best reflect the innovation of the paper. Hope you may understand.

  1. The English must be improved significantly. There are many grammatical errors throughout the manuscript.

Response: We are sorry for the spelling mistakes and grammatical errors caused by our carelessness. In the revised version, we have made significant efforts to remove the mistakes and errors and improved writing. We appreciate your elaborate efforts in reviewing. Thank you very much!

  1. Response to comment: Check carefully the correct way to write the unit. In addition, provide a space between the numerical value and unit.

Response: We are extremely grateful to you for pointing out this problem. We have gone over the units carefully and made a unified treatment for the expression of symbols in this paper. Spaces have been inserted between values and units. The above modified contents are reflected in the “Track Changes” mode in the article.

All the errors you picked and recommendations you proposed are greatly helpful for us to polish our manuscript. Thank you very much!

Best regards to you!

Corresponding author: Linlin Tan

Round 2

Reviewer 1 Report

I'm pleased to see the improvement of this paper. It can be published.

Reviewer 2 Report

The authors have sufficiently improved and corrected the manuscript.